# Contrasting nidification behaviors facilitate diversification and colonization of the Music frogs under a changing paleoclimate
Zhi-Tong Lyu [1,8], Zhao-Chi Zeng[1], Han Wan[1], Qin Li [2], Atsushi Tominaga[3], Kanto Nishikawa[4], Masafumi Matsui[5], Shi-Ze Li[6], Zhong-Wen Jiang [7], Yang Liu [1] ✉ & Ying-Yong Wang [1] ✉

In order to cope with the complexity and variability of the terrestrial environment, amphibians have developed a wide range of reproductive and parental behaviors. Nest building occurs in some anuran species as parental care. Species of the Music frog genus *Nidirana* are known for their unique courtship behavior and mud nesting in several congeners. However, the evolution of these frogs and their nidification behavior has yet to be studied. With phylogenomic and phylogeographic analyses based on a wide sampling of the genus, we find that *Nidirana* originated from central-southwestern China and the nidification behavior initially evolved at ca 19.3 Ma but subsequently lost in several descendants. Further population genomic analyses suggest that the nidification species have an older diversification and colonization history, while *N. adenopleura* complex congeners that do not exhibit nidification behavior have experienced a recent rapid radiation. The presence and loss of the nidification behavior in the Music frogs may be associated with paleoclimatic factors such as temperature and precipitation. This study highlights the nidification behavior as a key evolutionary innovation that has contributed to the diversification of an amphibian group under past climate changes.

Understanding how traits facilitate species diversification and adaptive radiation is a fundamental goal in evolutionary biology. The appearances of specific morphological characteristics or behaviors in organisms are usually related to adaptations to specific abiotic environments during evolutionary history and thus act as key innovations contributing to the diversification of life[1,2]. Nonetheless, most studies focus on understanding morphological or physiological traits that facilitate a species' survival abilities or exploitation on new resources, such as photosynthesis in cyanobacteria[3,4], wings in insects[5,6], and jaws in fishes[7,8]. Much less is known about key innovations

that facilitate organisms' reproduction and parental care, which in turn enhance their ability to adapt to climate change.

Amphibians, being the intermediate form of vertebrates that move from aquatic to terrestrial environments, play an important role in the evolution of tetrapods. The occurrences of four limbs and lungs enable the amphibian species to inhabit a wide variety of terrestrial habitats. However, their inability to produce amniotic eggs and their requirement for metamorphosis restrict them to close proximity with the freshwater environment, particularly during their vital reproductive periods. Thus, amphibians

[1]State Key Laboratory of Biocontrol, School of Ecology / School of Life Sciences, Sun Yat-sen University, Shenzhen 518107, China. [2]Zhejiang Tiantong Forest Ecosystem National Observation and Research Station, School of Ecological and Environmental Sciences, East China Normal University, Shanghai 200241, China. [3]Faculty of Education, University of the Ryukyus, Senbaru 1 Nishihara, Okinawa 903–0213, Japan. [4]Graduate School of Global Environmental Studies, Kyoto University, Yoshida-hon-machi, Sakyo-ku, Kyoto 606–8501, Japan. [5]Graduate School of Human and Environmental Studies, Kyoto University, Yoshida-Nihon-matsu, Sakyo-ku, Kyoto 606–8501, Japan. [6]Department of Food Science and Engineering, Moutai Institute, Renhuai 564500, China. [7]Key Laboratory of Animal Ecology and Conservation Biology, Institute of biology, Chinese Academy of Sciences, Beijing 100101, China. [8]Present address: CAS Key Laboratory of Mountain Ecological Restoration and Bioresource Utilization & Ecological Restoration and Biodiversity Conservation Key Laboratory of Sichuan Province, Chengdu Institute of Biology, Chinese Academy of Sciences, Chengdu 610040, China. ✉e-mail: liuy353@mail.sysu.edu.cn; wangyy@mail.sysu.edu.cn

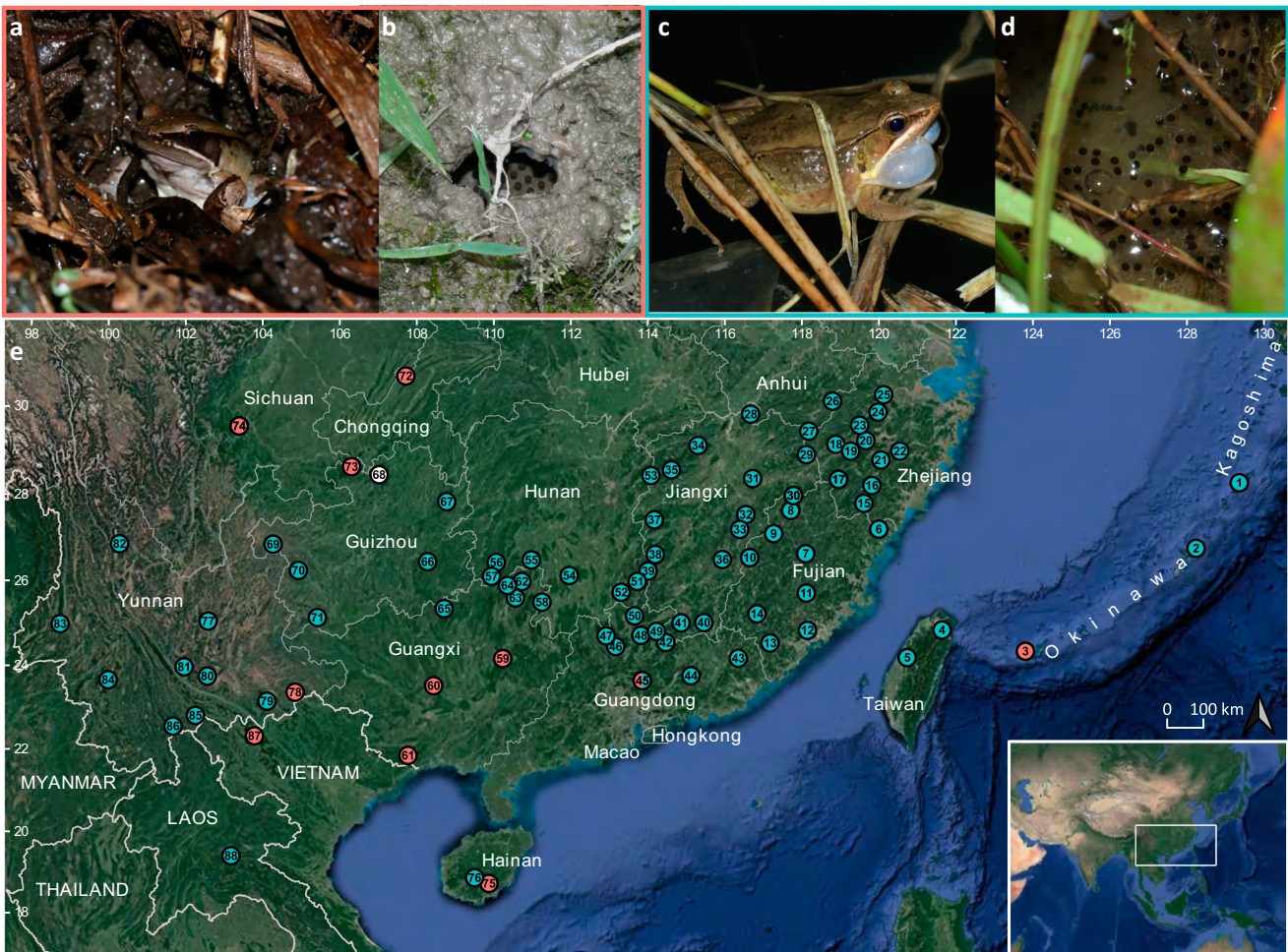

**Fig. 1 | Contrasting nidification behaviors of *Nidirana* and the sampling localities in this work. a** A male *N. guangxiensis* built a mud-based burrow as its nest and was calling inside it. **b** Eggs of *N. daunchina* in a mud nest. **c** A male *N. mangveni* called at the standing water surface. **d** Eggs of *N. guangdongensis* in the standing water surface. **e** Map showing sampling localities in this work. Numbers for localities are

corresponding to the information in Supplementary Table S1. Locality circles for species exhibiting nidification behavior, lacking nidification behavior, and nidification behavior remaining unknown, are colored with red, blue, and white, respectively.

are vulnerable to environmental changes, and hence any traits that facilitate their resilience to climate change or harsh environments, especially during breeding in terrestrial environments, would promote their diversification[9,10].

To overcome such environmental challenges and to safeguard their descendants during the fragile egg and larva stages, varied and even seemingly odd modes of incubating behaviors have evolved within anuran amphibians, such as gastric-brooding (*Rheobatrachus* spp.), mouthbrooding (*Rhinoderma darwinii*), and dorsal-brooding (*Pipa pipa*, *Gastrotheca* spp., etc.)[11–14]. Besides these, nest building (also known as nidification) is also regarded as an important strategy of parental care for anuran eggs, tadpoles, and juveniles[15–17]. Studies of nidification behavior usually focus on the most common foam nest type that occurs in many arboreal species (*Limnodynastes peronii*, *Rhacophorus* spp., etc.)[18,19], while mud nests are relatively overlooked as it has only been documented in a few lentic species, such as *Boana faber*[20], *Limnonectes limborgi*[21], and some species of *Nidirana*[22].

The Music frogs of the genus *Nidirana* are widespread in subtropical eastern and southeastern Asia, from the southernmost of the Ryukyu Archipelago, throughout southern China including Taiwan and Hainan islands, expanding to the northern Indochina Peninsula. They are known for their mud nest construction behavior which several congeners exhibit. This behavior can even be found in the generic name, as *Nidirana* is composed of Latin *nīdus* ("nest") and *rāna* ("frog")[22]. Among all of the known *Nidirana* species, ten congeners are reported to build mud nests for reproduction[23–28]. The male frogs will construct a refuse mud-based burrow

as their nest, call within it to attract females, and subsequently engage in mating within the nest. The eggs and larvae are, thus, protected in the mud nest from predators and external environmental changes (Fig. 1a, b). Compared with the arboreal foam nests, mud nests are usually constructed at the bank of lentic regions that face the complex challenges from both aquatic and terrestrial environments, and such nidification behavior may serve as an innovative adaptation to enhance their resilience. Meanwhile, other congeners of *Nidirana* do not possess such complicated behavior and just court and mate in standing water surfaces like most of other lentic frogs (Fig. 1c, d).

Considered as a defense mechanism in response to complicated habitats[14,29], we hypothesized that the presence and absence of mud nesting may be closely linked to environmental changes. In the distributional range of *Nidirana* across subtropical eastern and southeastern Asia, the emergence of a predominant monsoonal climate during the Cenozoic profoundly shaped terrestrial habitats and biogeography[30,31]. The East Asian Monsoon reached its peak intensity during the mid-Miocene, followed by a period of weakening until a re-strengthening during the Pliocene. Such fluctuations led to notable changes in the precipitation and temperature during this time frame[32–34], which might have played important roles in the evolution of nidification behavior in these lentic Music frogs. Furthermore, our previous studies and observations found that within *Nidirana*, species that display nidification behaviors are often highly endemic to narrow geographical areas, whereas congeners lacking nidification tend to be more widely

distributed[35,36]. Collectively, these pieces of evidence indicate that the nidification behavior of these species may be correlated with habitat specialties and unique colonization histories.

To understand the evolutionary history of the Music frogs and their unique mud nest construction behavior, we carried out a series of phylogenomic and population genomic analyses based on an extensive sampling from 18 of 20 described species of the genus. We hypothesized that the presence and absence of nidification behavior are associated with ancient climate changes, and that this behavior further acts as a key innovation contributing to the diversification and colonization of these frogs. Firstly, we employed genome-wide single-nucleotide polymorphisms (SNPs) to investigate their time-calibrated phylogeny and reconstruct their ancestral distributions. Secondly, to uncover the speciation of related species, we performed analyses on population genetics and demography for the congener subset with or without nidification behavior. Lastly, we examined the correlation between the nidification behavior and various associated bioclimatic factors.

## Results

### Prevalent discordance between mitochondrial and SNPs' phylogeny

Both ML and BI analyses for mitochondrial phylogeny generated nearly identical topologies (Supplementary Fig. S1a). The genus *Nidirana* was monophyletic, and four major clades A–D were revealed with strong support (BS ≥ 90, BPP = 1.00, respectively). The relationship among these four clades was (A, (B, (C, D))), even though the support for (C, D) were slightly weak in the ML tree. Each of the clades was further comprised of single or multiple lineages with strong support values (BS ≥ 99, BPP = 1.00, respectively). In total 18 lineages can be recognized, and 17 of them corresponded to 17 described species. The remaining unnamed lineage of the "Poyang population", which was previously recorded as *N. adenopleura*, was placed as a sister taxon to *N. mangveni*, thus this lineage was treated independently in the subsequent analyses. Furthermore, samples of *N.* "*guibeiensis*" clustered within *N. leishanensis*, indicating they are conspecific.

The phylogenetic result based on SNPs data (Fig. 2a; Supplementary Fig. S1b) showed a topology with several incongruences to the ones from the mitochondrial loci. The genus *Nidirana* was also recovered as monophyletic, but the interspecific relationships were not completely consistent. Clades A and B revealed from the mtDNA data were also supported by the SNPs data (BS = 100, respectively). However, taxa from mitochondrial clades C and D were intermixed and placed together in the phylogeny constructed from SNPs data (labeled as clade "C + D", BS = 100). This group can be further divided into four clades N1, N2, N3, and N4 (BS = 100, respectively). The relationship among these six clades in the SNPs phylogeny was ((A, B), (N1, (N2, (N3, N4)))). At the species level, most of the species, including the synonymy of *N.* "*guibeiensis*" with *N. leishanensis*, were well recognized by SNPs data (BS = 100, respectively). Yet there were two exceptions: species within clade N4 (the *N. adenopleura* complex) and clade A (the *N. pleuraden* complex). As illustrated by plots of the PCA result of SNPs data (Fig. 2b), all sampled individuals were primarily grouped into six clusters, corresponding to the above six clades. Congeners that possess nidification behavior (clades N1 and N2) were close to each other in phylogeny and PCA plots, despite the unknown nidification status for *N. yeae*.

### Spatiotemporal reconstruction of diversification patterns

The divergence time estimation from PAML indicated that the split between *Nidirana* and its sister taxon *Babina* was at ca 27.2 (95% CI 22.2–32.3) Ma (Supplementary Fig. S2). The Music frogs had the most recent common ancestor (MRCA) at ca 23.5 (95% CI 18.4–28.8) Ma. The divergence time for nidification species within clades N1 and N2 mainly occurred in the period of ca 10.0–15.4 Ma, while the divergence time for congeners without nidification behavior in clades A, N3, and N4 mainly occurred in the period of ca 2.1–9.3 Ma.

As inferred by the ancestral distribution reconstruction, it is highly likely that the MRCA of *Nidirana* inhabited the region encompassing the

Yunnan-Guizhou Plateau (Area C) and Luoxiao Mountain Range and nonboring river basin (Area E), and then began to migrate westward or eastward as two groups (Fig. 3a, b; Supplementary Fig. S3). The first group (clades A and B) mostly moved westward across the Yunnan-Guizhou Plateau and entered into the western Yunnan longitudinal valleys (clade A) and tropical hills in the northern Indochinese Peninsula (clade B). The second group (clade "C + D") primarily migrated eastward and subsequently split into three subgroups. One subgroup (clade N1) spread into different localities in Sichuan, Guangxi, Hainan, and Vietnam, where are close to the birthplace of the genus. Another subgroup (clade N2) migrated southeastward to Guangdong, with one population eventually dispersing to the Ryukyu Archipelago. The final subgroup (clades N3 and N4) evolved into the populations found in regions of central China (clade N3), as well as eastern mainland China and Taiwan Island (clade N4). Analyses on range shifts through time showed that the Music frogs had a major episode of dispersal events since Pliocene (Fig. 3c) and a minor episode of cladogenetic changes events around the mid-Miocene (Fig. 3d).

### Population structure and differentiation

The PCA result of the nidification species suggested six clusters (Fig. 4a), corresponding to six recognized species. Among them, clusters of *Nidirana nankunensis* and *N. okinavana* were relatively closer and forming a larger cluster, when compared with the other larger cluster including *N. chapaensis*, *N. hainanensis*, *N. yaoica*, and *N. daunchina*. This was consistent with their phylogenetic relationships (Fig. 2), and also reflected their different dispersal routes (Fig. 3b), which shaped their current distribution patterns (Supplementary Fig. S4a).

The result of the Admixture analysis suggested that the nidification species was comprised of six genetic populations (K = 6) (Fig. 4c). Almost each of them was distinct without genetic mixtures from other populations, except *Nidirana chapaensis* that was slightly intermixed with *N. hainanensis*. Genetic differentiations among populations of the nidification species were at a high level, with the lowest differentiation being observed between *N. yaoica* and *N. daunchina* ($F_{ST}$ = 0.329, $d_{xy}$ = 0.00193, respectively; Supplementary Fig. S4c).

For the *Nidirana adenopleura* complex without nidification behavior, the PCA plot showed that individuals were grouped into four clusters (Fig. 4b). Among them, samples of *N. guangdongensis* (GD), *N. mangveni* (MV), and the Poyang population (PY) were roughly separated from each other and formed three clusters. Surprisingly, samples of *N. adenopleura* were not gathered together but instead split into four populations, corresponding to their geographical distributions, respectively (Supplementary Fig. S4b): the samples from Taiwan Island (At) formed an independent cluster which was distinctly separated from others; samples from southern Zhejiang and northeastern Jiangxi (An) were gathered with MV; samples from northern Fujian and southeastern Jiangxi (Ac) were gathered with PY; and samples from southwestern Jiangxi (Aw) were gathered with GD. Therefore, the *N. adenopleura* complex was treated as seven populations in four clusters (At, An+MV, Ac+PY, and Aw+GD) in the subsequent analyses. Particularly, the rough relationship among clusters An+MV, Ac+PY, and Aw+GD in PCA result suggests their gradient of differentiation and admixture.

The Admixture analysis revealed that the *Nidirana adenopleura* complex was comprised of four genetic populations (K = 4) (Fig. 4d), consistent with the four clusters in the above PCA result. The insular population At was pure without genetic mixtures from other populations, while all of the other mainland populations were found to be somewhat intermixed. Samples within cluster An+MV had similar genetic mixture despite different proportions among individuals, and similar patterns also occurred within the other two clusters Ac+PY and Aw+GD. High genetic differentiation was found between population At and other populations (Supplementary Fig. S4d), which were more distinct in At vs. Aw ($F_{ST}$ = 0.265, $d_{xy}$ = 0.00168) and At vs. An ($F_{ST}$ = 0.177, $d_{xy}$ = 0.00148). Low genetic differentiation occurred within the clusters An+MV, Ac+PY, and Aw+GD ($F_{ST}$ < 0.044, $d_{xy}$ < 0.00107, respectively).

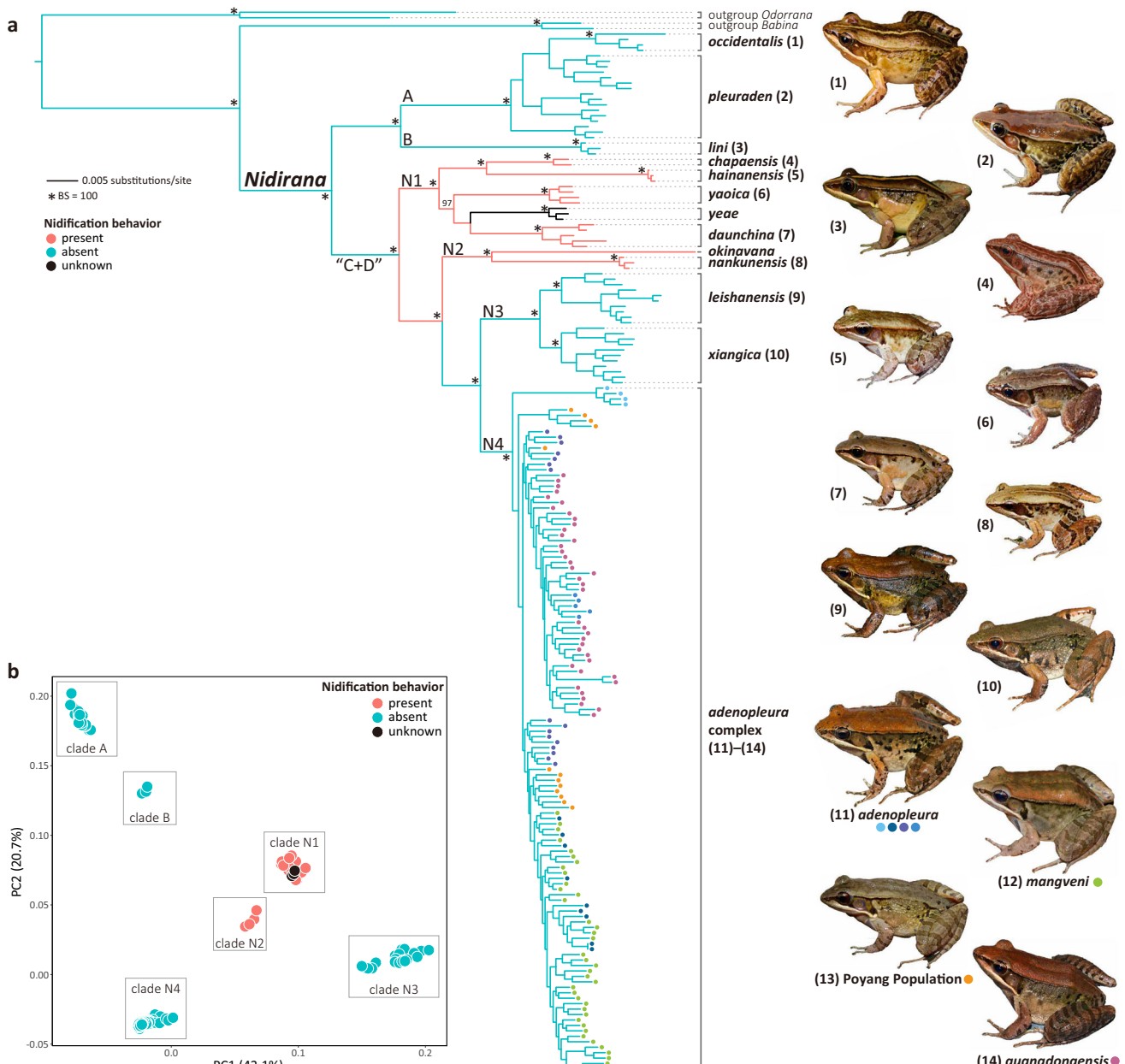

**Fig. 2 | Phylogeny and PCA relationship for genus *Nidirana* based on SNPs data. a** Phylogeny of the genus *Nidirana*, using maximum likelihood based on SNPs data. Values of bootstrap supports (BS) larger than 90 are shown. Within the *N. adenopleura* complex, different populations are labelled with solid circles of different colors: *N. mangveni*, the Poyang population, *N. guangdongensis*, and four populations of *N. adenopleura*: Taiwan, north, center, and west. **b** Two-dimensional PCA plot for genetic variation in the genus *Nidirana*.

## Demographic trajectories and dynamics

The nidification species have a relatively plain demographic history, as can be seen from the demographic trajectories depicted at ca 5–10 Ma with the effective population size showing a distinct decrease (Fig. 4e). After this single dramatic decrease, no other distinct fluctuation could be observed for the effective population size of each of the nidification species until the present. *Nidirana okinavana* was not included in this analysis due to a limited sample size. By contrast, in the analysis of demographic trajectories from ca 5 Ma, the effective population size gradually decreased in each of the seven populations within the *N. adenopleura* complex (Fig. 4f). Even though the degrees of decrease and processes varied among different populations, no distinct bottlenecks or population expansion events were inferred. For demographic dynamics analysis for the *Nidirana adenopleura* complex, we selected three pairs of populations based on the upstream results (Figs. 2, 4; Supplementary Fig. 1): (1) An vs. MV, representing the populations that showed distant relationships in mtDNA tree but were in one cluster using SNPs data; (2) An vs. Ac, representing the populations that have been clustered into a single lineage in mtDNA tree but were separated distinctly in SNPs data; (3) At vs. MV, representing the populations that were placed distinctly using both datasets. The results revealed that reciprocal migrations were present between the populations pairs of An and MV, nonreciprocal migration was present from population Ac to population An, and migration was absent between the populations of At and MV (Supplementary Table S3).

## Environmental correlation of nidification behavior

The GLMMs analysis indicated that the presence/absence of nidification behavior of Music frogs is markedly associated with 12 bioclimatic variables ($P$-values < 0.05) and especially three variables with $P$ values < 0.01 (Supplementary Table S4). The six major variables were annual temperature

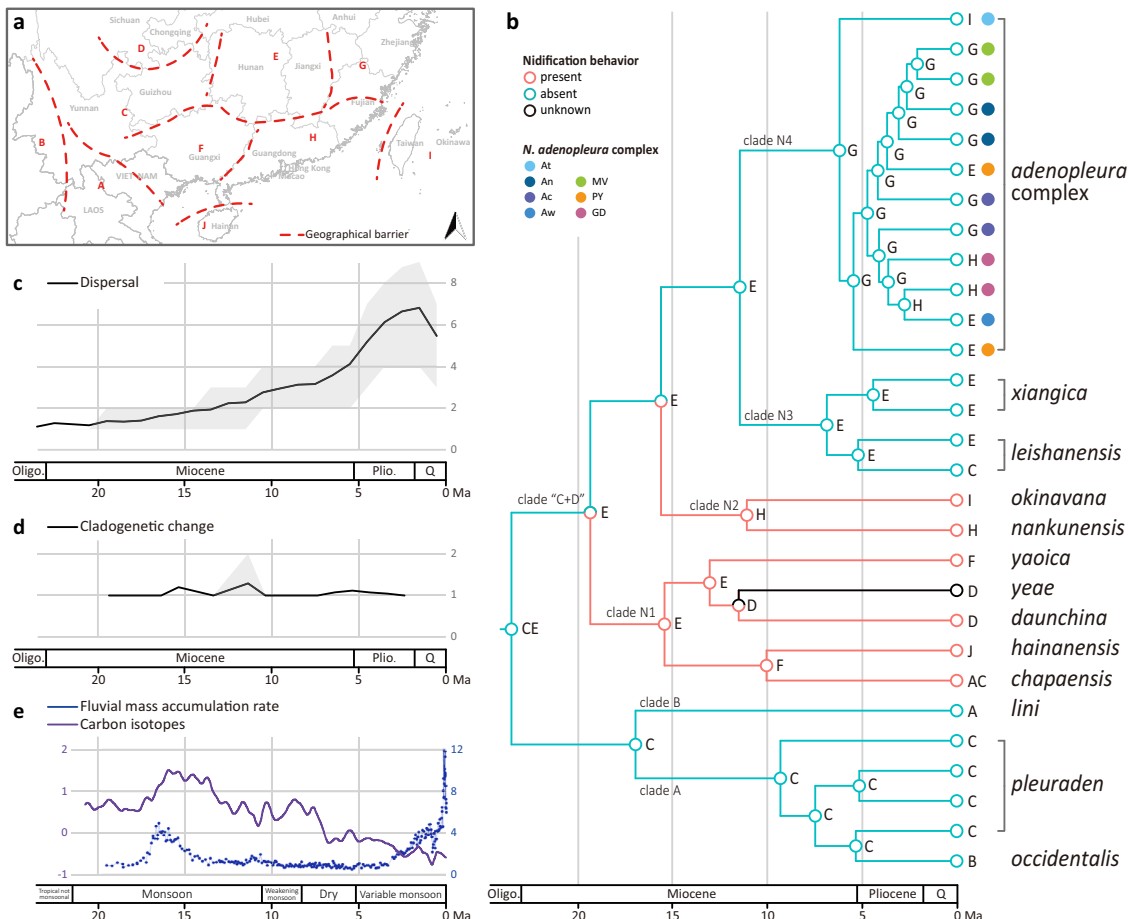

**Fig. 3 | Spatiotemporal reconstruction of *Nidirana*. a** Ten geographical areas delineated for the genus *Nidirana*. Characters for geographical areas and dashed lines for barriers are corresponding to the information in Supplementary Table S2. **b** Ancestral distributions reconstructed by Dispersal-Extinction-Cladogenesis model. Areas with the highest probabilities at nodes are shown with the same characters in Fig. 3a. Within the *N. adenopleura* complex, populations are labelled with solid circles of different colors: *N. mangveni* (MV), the Poyang population (PY),

*N. guangdongensis* (GD), and four populations of *N. adenopleura*: Taiwan (At), north (An), center (Ac), and west (Aw). **c** Dispersal events along branches through time. **d** Cladogenetic change events through time. **e** Development of East Asian Monsoon from Miocene with the fluctuations of fluvial mass accumulation rate representing level of precipitation and carbon isotopes representing level of temperature[30,34,45].

range (Bio 7), temperature seasonality (Bio 4), precipitation seasonality (Bio 15), precipitation of warmest quarter (Bio 18), precipitation of coldest quarter (Bio 19), and minimum temperature of coldest month (Bio 6) (Fig. 5). As distinctly indicated in the boxplots and scatterplots, *Nidirana* species with nidification behavior are more likely to inhabit environments with smaller fluctuations in temperature (Bio7 and Bio4), warmer weather in the coldest month (Bio6), and higher precipitation of the warmest quarter but lower precipitation of the coldest quarter (Bio18, Bio 15, and Bio19), while the species without nidification behavior were adapted to the opposite conditions.

## Discussion
### Single origin of nidification behavior
Previous phylogenetic studies using mitochondrial markers suggested the nidification behavior may evolve multiple times independently[22,24]. In this work, the mitochondrial phylogenetics with complete sample coverage of genus yielded the same result as previous studies. However, when a larger dataset of SNPs was used, the phylogenomic analyses revealed that the *Nidirana* species with nidification behavior essentially form two paraphyletic clades N1 and N2, and these two clades and another two clades N3 and N4, which are species without the nidification behavior, have a MRCA to form the clade "C + D" (Fig. 2). Populations within this "C + D" clade have a similar spatiotemporal history that represent the eastward colonizers (Fig. 3b).

Nidification is an obviously complicated behavior as it may be involved in the courtship and the mating site, and consists of different steps: building the burrow, emitting calls, and mating[15,37,38], which can be reasonably expected to be of a single origin rather than evolving multiple times independently during their evolutionary history[39,40]. Furthermore, the above-mentioned steps of nidification are primarily initiated and performed by the male individuals, thus maternal mitochondrial data might take bias for the origin of this behavior. Hence, we consider that the behavior of nest construction is much more likely to have initially evolved at the MRCA of clade "C + D" (ca 19.3 Ma), and was subsequently lost at the MRCA of clades N3 and N4 (ca 11.4 Ma).

### Different evolutionary histories of the congeners with different behavior
The nidification species have relatively consistent phylogenetic relationships in both analyses. Population genomics suggest that no sign of admixture and high levels of genetic differentiations among them. Demographic analyses further indicate their relatively stable population size and much plainer demographic history. As revealed from the spatiotemporal reconstruction, divergence time for these species mainly concentrated around the mid-Miocene, accompanying with a minor episode of cladogenetic changes of range evolution in this period. Thus, the nidification species are considered to have an older history of diversification and colonization.

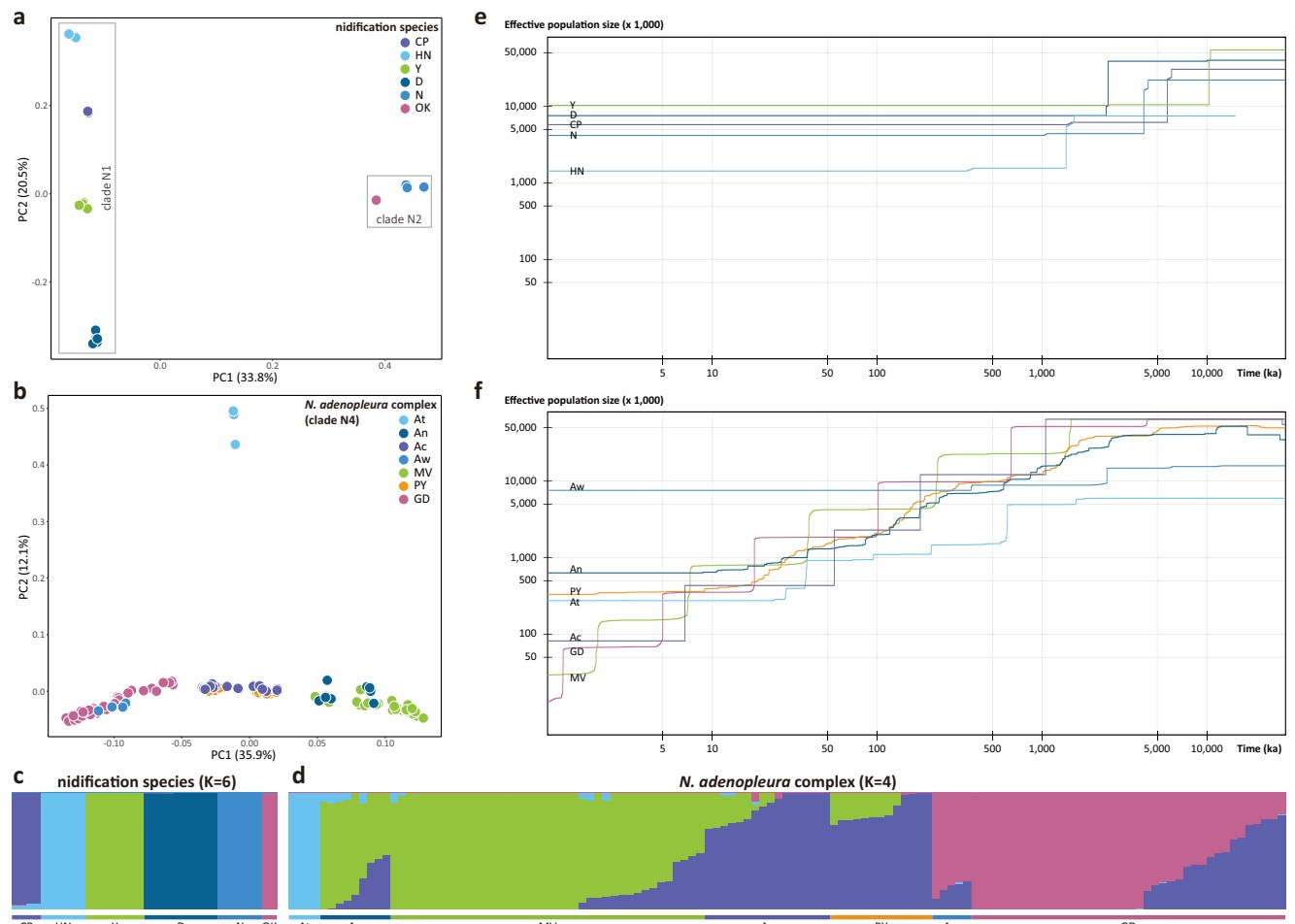

**Fig. 4 | Population structure and differentiation. a** PCA plot of PC1 and PC2 for nidification species. **b** PCA plot of PC1 and PC2 for the *Nidirana adenopleura* complex. **c** Admixture plot for nidification species with K = 6. **d** Admixture plot for the *N. adenopleura* complex with K = 4. **e** Demographic trajectories for different populations for five nidification species. *Nidirana okinavana* was not included in this analysis due to a limited sample size. **f** Demographic trajectories for different populations for seven populations of the *N. adenopleura* complex. Labels for different populations: *N. chapaensis* (CP), *N. hainanensis* (HN), *N. yaoica* (Y), *N. daunchina* (D), *N. nankunensis* (N), *N. okinavana* (OK), *N. mangveni* (MV), the Poyang population (PY), *N. guangdongensis* (GD), and four populations of *N. adenopleura*: Taiwan (At), north (An), center (Ac), and west (Aw).

By contrast, the *Nidirana adenopleura* complex without nidification behavior may have experienced a recent rapid radiation. There is discordance in the topologies derived from mtDNA and SNPs data for the complex, with four clusters revealed by SNPs data being different from the four lineages identified in the mitochondrial phylogeny. Several factors may drive such patterns, such as incomplete lineage sorting from ancestral polymorphism and introgressive hybridization between taxa[41–43]. Combining the results of demographic analyses, it is most likely that incomplete lineage sorting from ancestral polymorphism has driven these incongruent phylogenetic topologies. The decreasing effective population sizes without bottlenecks and population expansion for different populations indicate their isolations from each other, and secondary contacts are unlikely to exist. Besides, the migrations mostly occurred between populations with a closer SNPs relationship (An vs. MV), and became fewer (An vs. Ac) or even absent (At vs. MV) as the divergence increased, suggesting low likelihood of introgressive hybridization.

Moreover, similar to the condition of the *Nidirana adenopleura* complex, conflicting topologies are also observed in terminal lineages within clade A (namely *N. pleuraden* complex), and complicated intraspecific relationships are found within clade N3 (*N. leishanensis* and *N. xiangica*) (Fig. 2; Supplementary Fig. S1). All of these taxa are without nidification behavior, and their major divergent time is relatively close from each other (Supplementary Fig. S2). A recent rapid radiative evolution similar to that in

the *N. adenopleura* complex could be also expected in these species, which is closely associated with the major episode of dispersal events since Pliocene.

## Paleoclimate changes benefit *Nidirana* congeners with different behaviors

Environmental correlation analysis illustrates that nidification species tend to inhabit environments where the temperature is relatively stable and warmer and the precipitation is mainly concentrated in the warmest quarter, which are the typical characteristics of strengthened monsoon climate. However, the congeners without nidification behavior are adapted to the opposite conditions. Such contrasting tendencies suggest that the presence or absence of nidification behaviors are associated with variations in temperature and precipitation due to climate changes.

Through the ancient development stages of monsoons in eastern Asia (Fig. 3e), the East Asian Monsoon intensified to a maximum during the mid-Miocene. This resulted in an environment characterized by extremely high precipitation and stable temperature in southern China, which in turn facilitated the diversification of Music frogs exhibiting nidification. Since Pliocene, a recent rapid radiation occurred in *Nidirana*, generating the diversity of extant congeners lacking nidification behavior. During this time period, the East Asian Monsoon was weakening, resulting in a dryer and cooler climate.

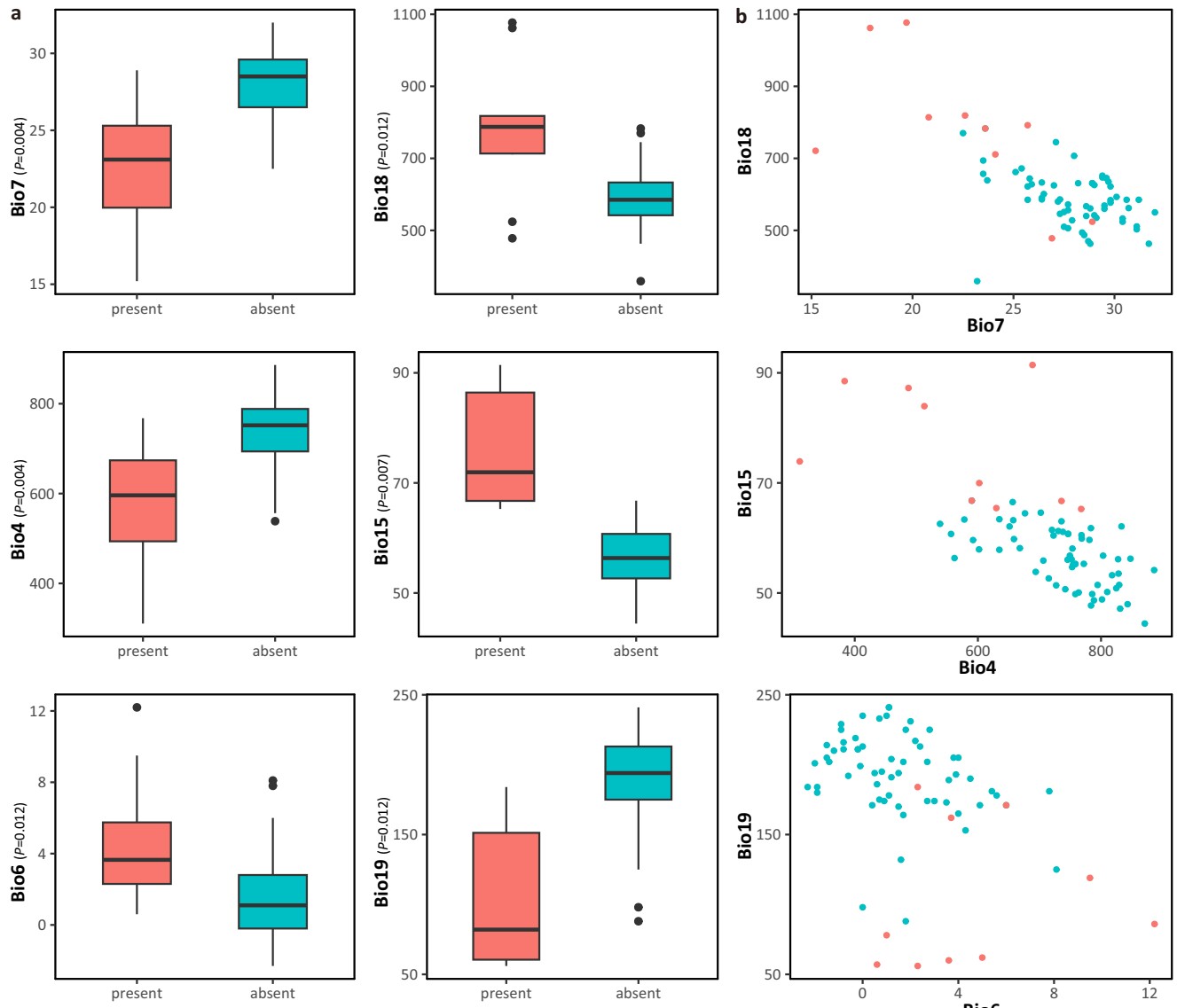

**Fig. 5 | Environmental correlation of nidification behavior. a** Boxplots showing the correlation for absence/presence of nidification behavior with Bio 7 (annual temperature range), Bio 4 (temperature seasonality), Bio 6 (minimum temperature of coldest month), Bio 18 (precipitation of warmest quarter), Bio 15 (precipitation seasonality), and Bio19 (precipitation of coldest quarter). **b** Scatterplot showing the associations for absence/presence of nidification behavior with Bio 7 and Bio 18, with Bio4 and Bio15, and with Bio6 and Bio19.

The nidification behavior is revealed to occur before the mid-Miocene and the maximum of the East Asian Monsoon (Fig. 3b). During the intensification of monsoons in southern China, the constantly higher temperatures and higher levels of precipitation caused an increase of overland floods (Fig. 3e)[44,45]. For frogs inhabiting open lentic ecosystems such as ponds, swamps, and paddy fields[23,26,28], constructing nests for mating and incubating eggs would act as a means to mitigate the impact caused by the perennial high temperatures, heavy rainfall, and frequent floods. Such nests would also be an advantage against desiccation after flood waters recede, especially during their breeding season in summer. Therefore, the frog group benefited from the nidification behavior in such climate and underwent species diversification. Since ca 8 Ma, the temperature cooled down and the East Asian Monsoon substantially weakened, resulting in decreased precipitation and overland floods (Fig. 3e)[32]. Consequently, lacustrine environments were well developed, forming abundant lentic ecosystems[46]. As a result, the Music frog congeners without nidification behavior presumably started their rapid radiation from that time onwards

till the present since they were not constrained by the need for nesting during mating.

### *Nidirana* possesses adaptative flexibility under climate changes

A nest is an animal architecture that is usually elaborate and painstakingly built to protect offspring from predation and environmental variables[47,48]. While nests are primarily associated with birds of various types[47], such architecture can be found across vertebrate taxa, including the little-known nests constructed by anuran amphibians[16,49].

Our work investigates the evolutionary history of the Music frogs and their unique mud nest construction behavior, which has received less attention in previous studies. These findings indicate that mud nests likely evolve as a key evolutionary innovation during the paleoclimate characterized by perennial high temperatures and heavy rainfall. However, it was subsequently lost in certain descendants as the environment transitioned to a dryer and colder state. In contrast to the nesting behavior observed in Music frogs, the foam nest in the Treefrog family Rhacophoridae has been

found to be well-adapted for cold and dry settings[18,50]. Foam nests are usually built in trees and are regarded as one of the adaptive mechanisms for arboreal life, being a shelter against the dry air that is uneasy to be replaced[18,50]. On the contrary, the mud nests built on the bank of lentic regions serve as a residence during intense rainfall and floods and can be easily discarded when no longer needed. Therefore, the construction behavior of mud nests in *Nidirana* would exhibit greater evolutionary flexibility in response to environmental variations. Climate models have predicted a significant increase in temperature in the near future, accompanied with intensified heavy rainfall and flooding[51,52]. Given their evolutionary history, it is plausible that the nesting behavior could resurface in *Nidirana* as a strategy to endure the increasingly hot and humid climate and flourish in the future. However, it is claimed that climate changes during the Anthropocene are characterized by greater variability and an increase in extreme weather events[53,54]. Thus, the extent of Music frogs' adaptability in the face of unpredictable future climate conditions remains undetermined.

Given its intricate nature, it is imperative to conduct further research to unveil the genetic basis of the nidification behavior. The regulatory mechanisms underlying this complicated behavior at genetic, physiological, and behavioral levels are not well understood, particularly regarding the transition between nidification and non-nidification. *Nidirana* congeners show great potential for studies in this field. Our present study sheds light on the external environmental forces, and further research will be conducted to clarify and comprehend the adaptive mechanisms when tetrapods evolved from water to the land.

## Methods

### Sampling and sequencing

Field surveys were conducted from 2014 to 2021, throughout the distribution range of the genus *Nidirana* (Fig. 1e; Supplementary Table S1). A total of 237 samples of 18 *Nidirana* from 83 localities were collected. Four additional samples of *Babina* and *Odorrana* species were collected as outgroups[26]. All specimens were fixed in 10% buffered formalin and later transferred to 70% ethanol; muscle or liver samples were obtained from euthanized specimens and then preserved in 95% ethanol and stored at −40 °C in the Museum of Biology, Sun Yat-sen University. All the procedures related animals were performed in accordance with the ethical guidelines and approval of the of Institutional Animal Care and Use Committee of Sun Yat-sen University (2005DKA21403-JK).

Genomic DNA was extracted using a CTAB-extraction protocol with Proteinase K to digest tissue. All the DNA integrities were inspected on a 1% agarose gel to ensure the completion of DNA. All collected samples, except two of *Babina* species due to the quantity limitation, were amplified and sequenced for two mitochondrion segments (mtDNA), namely partial 16 S ribosomal RNA gene (*16 S*) and partial cytochrome C oxidase 1 gene (*COI*). Primers, PCR amplification, and sequencing protocol followed our previous studies[55–57]. In addition, published sequences from 22 individuals (13 with both *16 S* and *COI* data and nine with only *16 S*; Supplementary Table S1) of *Nidirana* and *Babina* species were downloaded from GenBank and incorporated into our dataset for mitochondrial phylogenetic analysis.

Double-digest restriction site-associated DNA sequencing (ddRAD-seq)[58] conducted by Shenzhen RealOmics (Biotech) Co., Ltd. was applied to a subset consisting of 192 samples of 16 *Nidirana* species and four samples of the out-groups (Supplementary Table S1). Restriction enzymes EcoRI and NlaIII were used for double digests. The enzyme digestion products were ligated to index adaptors for PCR-amplification. Paired-end 150-bp reads were generated on an Illumina Hiseq platform. Sequence data from ddRAD-seq were performed for de novo SNPs calling using Stacks v2.53[59]. Raw sequences were filtered using process_radtags tool, and reads with Phred scores lower than 20 were discarded and the rest were all trimmed to 130 bp. After that, the reads were assembled and SNPs were called based on the main Stacks pipeline: ustacks, cstacks, sstacks, gstacks, and populations. To ensure sufficient yields of polymorphic loci and exclude erroneous assemblies, we tested different combinations of parameters, and the final parameter set was: minimum depth of coverage required to create a stack

(m) = 3, maximum mismatches allowed between stacks (M) = 2, and maximum allowed mismatches between loci when building the catalog (n) = 2. VCFtools v0.1.16 was used for quality filtering[60], and the final filtered SNPs dataset of 196 individuals and 1,460,825 polymorphic sites was finally obtained for downstream analysis.

### Phylogenetic analysis

For mitochondrial phylogeny, raw sequences of *16 S* and *COI* were aligned respectively by the Clustal W algorithm with default parameters and subsequently refined in MEGA 6[61,62]. The two segments, 1,045 base pairs (bp) of *16 S* and 639 bp of *COI*, were concatenated into a 1,684-bp matrix. The matrix was tested in jmodeltest v2.1.7[63], resulting in the best-fitting nucleotide substitution model as GTR + I + G based on the Bayesian information criterion. The dataset was analyzed using maximum likelihood (ML) in RAxML v8.0[64], and Bayesian inference (BI) in MrBayes 3.2.4[65]. In the ML analysis, the bootstrap consensus tree inferred from 1,000 replicates was used to represent the evolutionary history of the taxa analyzed. Two independent runs were conducted for the BI analysis, each for 10,000,000 generations with four Markov Chains Monte Carlo chains and sampled every 1,000 generations, and the first 25% of samples discarded as burn-in. The result was confirmed to be converged based on the average standard deviation of split frequencies below 0.01, as well as effective sample size larger than 200 which was evaluated in Tracer v1.7[66]. ML bootstrap support (BS) larger than 90 and Bayesian posterior probabilities (BPP) larger than 0.95 were considered to be strong supports.

For filtered SNPs data, vcf2phylip (https://github.com/edgardomortiz/vcf2phylip, last accessed November 20, 2021) was used to generate the shared SNPs matrix. The matrix was analyzed using ML in RAxML-NG[67]. The best-fitting nucleotide substitution model tested by jmodeltest was GTR + I + G based on BIC. Branch supports were evaluated with 1,000 rapid bootstrapping replicates, and BS larger than 90% were considered to be strong supports. Furthermore, the filtered SNPs matrix was extracted by VCFtools to retain 192 *Nidirana* individuals. PLINK 1.9[68] was used to calculate linkage disequilibrium and prune the SNPs matrix to those with linkage disequilibrium < 0.2. Principle component analysis (PCA) was conducted in PLINK to find the best low-dimensional representation of each SNP in the genus *Nidirana*.

### Divergence time estimation

The divergence time for the genus Nidirana was estimated using the MCMCTREE program in PAML v4.8[69]. The ML topology based on the SNPs data was used as the reference tree. Due to the lack of fossils for *Nidirana* and its related genera, an inference time from the previous study was used to calibrate the clock, and the root age was set to 28.63 Ma with the 95% credible interval (CI) as 24.0–33.0 Ma[70]. Data binning strategy was used to reduce sampling error that divided the SNPs data into five bins[71,72]. The ML estimates of branch lengths for the bins of alignments and the substitution rate per time unit were obtained by the BASEML program in PAML. In MCMCTREE, the clock model was set as independent rate model. The Markov Chains Monte Carlo run was first executed for 10,000,000 generations as burn-in and then sampled every 150 generations until a total of 100,000 samples were collected. Two independent runs using random seeds were performed to examine whether similar results were obtained.

### Ancestral distribution reconstruction

To infer the ancestral distributions of *Nidirana*, we applied the Dispersal-Extinction-Cladogenesis model in a Bayesian framework implemented in RevBayes[73,74]. The trees dataset was estimated using BEAST v2.5 for a total of 10,000 trees[75], and the phylogeny with divergence time estimating from PAML was used as the condensed tree, with out-groups removed. For the distributions statement, we divided the distribution of the whole genus into ten areas (A–J) on the basis of topography, which were separate from each other by distinct geographical barriers of mountains, rivers, or straits (Supplementary Table S2). Then tip lineages were assigned areas based on

their known occurrences. The dispersal rates between areas were scaled to their relative distances, represented by 10^ the minimum number of boundaries needed to be crossed from one area to the other. We used a time-stratified model with the time split into two epochs by the emergence of Taiwan island (the time bounds of this breakpoint: 9–6 Ma)[76], which means that before this break there was no connectivity between area I and other nine areas, whereas after this break all ten areas are allowed to be connected. Two cladogenetic event types, allopatric and subset sympatric speciation were considered in the Dispersal-Extinction-Cladogenesis model, and the maximum number of areas was limited to two, given narrow distributions across extant lineages and limited dispersal ability in this group. We performed the analyses in RevBayes for 100,000 generations, and all other parameters were set to the default following the tutorial. With Markov Chains Monte Carlo samples, we annotated the condensed tree with ancestral states and generated stochastic mapping across the phylogeny to count biogeographic events over time. All parameter posteriors have effective sample sizes well over 1000. Meanwhile, we applied stochastic character mapping to generate simulated histories consistent with the observed data. With 1000 stochastic maps, we discarded the first 25% as burn-in, and parsed sequences of range states along the branches (anagenetic changes) and at nodes (cladogenetic changes) in the phylogeny. We then estimated numbers of range shifts through time by slicing the phylogeny into 1-Ma time windows, mainly focusing on dispersal/range expansion and cladogenetic changes (including allopatric and subset sympatric speciation).

### Population and demographic analyses
Detailed population and demographic analyses were performed on the *Nidirana* congeners with or without nidification behavior, respectively. *Nidirana chapaensis*, *N. hainanensis*, *N. yaoica*, *N. daunchina*, *N. nankunensis*, and *N. okinavana* were used as nidification species. *Nidirana adenopleura* complex (including populations of *N. adenopleura*, *N. guangdongensis*, and *N. mangveni*) are used as the representatives for the congeners without nest construction behavior.

The SNPs dataset of nidification species including 18 samples and dataset of the *Nidirana adenopleura* complex including 127 samples, were extracted from the original filtered SNPs matrix by VCFtools and further pruned with linkage disequilibrium < 0.2. PCA was conducted in PLINK for the SNPs dataset of the nidification species and *N. adenopleura* complex, respectively. The population structure among individuals of the complex was inferred by Admixture 1.3[77] under different numbers of subgroups (K) setting from 2 to 10. The most likely number of genetic clusters was computed with a 10-fold cross-validation error. To obtain the level of genetic differentiation among species, Wright's fixation index ($F_{ST}$) and absolute genetic divergences ($d_{xy}$) statistics were computed using the populations program in Stacks.

To understand effective population size changes through historical time, demographic trajectories for each of the nidification species and the *Nidirana adenopleura* complex were inferred using Stairway Plot 2[78]. The 1D folded observed site frequency spectrum for each population was calculated on the SNPs matrix using easySFS (https://github.com/isaacovercast/easySFS, last accessed October 28, 2021), using a strategy of maximizing the number of segregating sites. In the Stairway Plot 2 analysis, the mutation rate per site per generation was set as 7.76e-10[79], the generation time was set at 1 years, and 200 bootstrap iterations were conducted.

The upstream analysis results showed that species within the *Nidirana adenopleura* complex have relatively complicated evolutionary histories (see Results above), therefore, we used fastsimcoal v2.7 to further infer the demographic dynamics[80]. To simplify its demographic histories, we performed four scenarios for three pairs of *Nidirana* populations selected according to the results of upstream analyses (Supplementary Table S4). The four scenarios were: (1) mig0: the absence of migrations between the two populations; (2) mig12: the presence of nonreciprocal migration from population 1 to population 2; (3) mig21: the presence of nonreciprocal migration from population 2 to

population 1; (4) mig2: the presence of reciprocal migrations between the two populations. The population pairs were selected based on the results of upstream analyses. The 2D folded observed site frequency spectrum for population pairs was also calculated using easySFS, and the mutation rate per site per generation was also set as 7.76e-10[79]. To maximize the likelihood of each model, we ran 100 expectation-conditional maximization cycles with a total of 100,000,000 coalescent simulations. For each model, we obtained the best likelihood values and estimated parameters from 100 optimizations. The best model was determined using the Akaike information criterion weight method[81].

### Environmental correlation of nidification behavior
To assess the correlation and dependence of the nidification behavior with the environmental factors, we performed Generalized Linear Mixed Models (GLMMs) using the *glm* function in the *lme4* R package with a binomial distribution and a logit link[82]. For environmental factors of each sampling locality (Supplementary Table S1) of *Nidirana* species, we extracted 19 bioclimatic variables (Supplementary Table S4) from WorldClim v2.1 at a resolution of $1 \times 1$ km for each variable[83]. P-value in the result was adjusted using False Discovery Rate method. Boxplots and scatterplots were visualized with the *ggplot2* R package[84].

### Reporting summary
Further information on research design is available in the Nature Portfolio Reporting Summary linked to this article.

### Data availability
All mitochondrial DNA Sanger sequencing data were uploaded to NCBI Nucleotide Database and the accession numbers are provided in Supplementary Table S1. All raw sequencing data generated by ddRAD-seq were uploaded to NCBI Sequence Read Archive under the BioProject accession number PRJNA1091219. Alignment for phylogeny and supplementary tables and figures are available on Dryad at https://doi.org/10.5061/dryad.wstqjq2rs.

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

## Acknowledgements

We thank Jian Wang, Yao Li, Guoling Chen, Bicheng Zhu, Xiao-Yun Wang, Si-Min Lin, Yun-Ming Mo, Song Huang, Zhongxing Wang, Jian Zhao, Hai-Long He, Jian Luo, Jian-Huan Yang, and Ying Liu for their help in the field and lab. We thank Zhiyong Yuan and Jin-Long Ren for providing samples even though they were not used in this study. Richard H. Ree and Michael J. Landis provided guidance on the RevBayes analysis. Isaac Overcast, Mengjie Jin, He Chen, and Xin-Kai Wu provided valuable comments on the initial manuscript. This work was supported by DFGP Project of Fauna of Guangdong-202115 and the Interdisciplinary Innovation Team of the Chinese Academy of Sciences (CAS) "Light of West China" Program (xbzg-zdsys-202207).

## Author contributions

ZTL and YYW conceived the study; ZTL, ZCZ, and HW collected samples and conducted laboratory work; A Tominaga, K Nishikawa, M Matsui, and SZ Li, contributed materials; ZT Lyu, ZC Zeng, Q Li, and ZW Jiang analyzed the data; ZT Lyu, ZC Zeng, Q Li, and YL wrote the manuscript with supports from all authors.

## Competing interests

The authors declare no competing interests.
