## [Peer review file · Communications Biology]

Reviewers' comments:

Reviewer #1 (Remarks to the Author):

In general, I am impressed with the study design presented by the authors in this manuscript. Amphibians are vulnerable to environmental changes. Consequently, any traits that facilitate their resilience to climate changes (e.g., breeding in terrestrial environments) would promote their diversification. In this manuscript the authors investigated whether nidification behaviour of frogs genus *Nidirana* facilitate their diversification and colonisation under a changing paleoclimate. They employed a series of phylogenomics and population genomics analyses for 18 of 20 currently recognised species of *Nidirana*. Their study suggested that nidification behaviour in this genus initially evolved at approx. 19.3 Mya and eventually lost in several descendants, which might be linked to paleoclimatic factors such as temperature and precipitation. This study shed light to our current knowledge on the importance of understanding reproductive behaviour (i.e., nidification) of amphibians, which is not commonly studied to date, as key innovation that contribute to their diversification (i.e., genus *Nidirana*) under past climate change.

The authors described and wrote their study clearly and in nice story flow, which made it easy to understand while reading it. While I don't have major comments to improve the quality of the manuscript, I would like to give some minor comments/suggestions, as follow:

Title: I suggest adding "of the genus *Nidirana*" after "amphibian group" to emphasise that this study is done for the genus *Nidirana*. This would improve the clarity that the study is done for the genus *Nidirana* and that the findings might not apply for other species/genus.

Page 3 line 63. Please replace "nest building" with "nest building (nidification)" to introduce the term for the first time in the text. This would facilitate the reader who heard the term nidification for the first time to learn about this term.

Page 5 line 99. Please replace "all described species of the genus" with "18 of 20 described species of the genus". The two newly described species (*Nidirana chongqinensis* and *N. noadihing*) were not included in this study. I understand that by the time the author started their study, these two species were not described yet. Therefore the authors didn't include these two species. However, since there has been two additional described species, updating the sentence in the text would be recommended.

Please be consistent in labelling the samples used in this study. The authors wrote *Nidirana guibeiensis* throughout the text but in the Supplementary Table S1 *Nidirana* "guibeiensis".

While I understand that it might not be the scope of this manuscript, I am wondering if it is possible to somehow elaborate/indicate in one or more sentence(s) how the results from this study could be used to project/estimate future populations of this genus under the climatic factors defined in this study.

Reviewer #2 (Remarks to the Author):

The study “Contrasting nidification behaviors facilitate diversification and colonization of an amphibian group under a changing paleoclimate” deals with the frog genus *Nidirana*, that comprises species that either did or did not evolve parental care behaviors in form of building mud nests (here called nidification behaviors). The study is overall well written, the methods are sound and the results are presented and (for the most part) discussed nicely.

General comments

For this study a large amount of *Nidirana* specimens were collected from a large range of localities in over 7 years and their DNA samples were used for different comparative phylogenetic analyses and PCAs, with the overall goal to study the evolution of nidification behaviors in the context of historical climate change (using ancestral distribution reconstruction and correlating environmental factors with parental care behaviors). All in all, the methods used in this context are convincing, leading to strong results, which are presented in nice figures (both in the manuscript and in addition in the supplementary materials). Also the introduction presenting the overall topic and giving a rationale for conducting this study is very well written. The only section that in my opinion needs some strong rewriting is the discussion: while the different aspects of the study are nicely discussed under the three headings, I am missing a general discussion of the overall results in context with other studies, such as comparisons to other animal groups, other behaviours/features evolved due to climate changes etc. I am furthermore missing a general conclusion at the end of the discussion and maybe even a statement regarding the meaning of the results in the lights of current climate change.

Minor comments

- line 33-36: this sentence is very hard to understand and should be rewritten in a more comprehensible way
- line 59-64: most of the amphibian parental care reviews cited here are rather old. The field has developed substantially and there are several much more recent reviews about this topic
- line 67-68: please provide sources (or a review) for the documentation of mud nests (at least in Boana and *Limnonectes*)
- line 456: species name italic
- line 479: please cite the creators of the ggplot2 package

Dear reviewers,

On behalf of the co-authors, we are very grateful to your constructive suggestions on our work entitled “**Contrasting nidification behaviors facilitate diversification and colonization of an amphibian group under a changing paleoclimate**” (COMMSBIO-23-4560). These suggestions greatly help us to improve our manuscript.

We have therefore reviewed your comments carefully and tried our best to revise our manuscript accordingly.

Please find below for a point-by-point response to the reviewers’ comments and concerns. The reviewers’ comments are presented in italicized font and specific concerns have been numbered. Each comment is followed by our response. Moreover, all changes in the latest manuscript have been made using Microsoft Word’s Track changes feature, as well as a clean version with marks provided.

We thank you again for giving us an opportunity to revise this manuscript and look forward to hearing from you.

Thanks once for your time!

Sincerely,

Prof. Yang Liu, on behalf of all coauthors

Responses to the comments of Reviewer #1

Comment 1. Title: I suggest adding “of the genus *Nidirana*” after “amphibian group” to emphasise that this study is done for the genus *Nidirana*. This would improve the clarity that the study is done for the genus *Nidirana* and that the findings might not apply for other species/genus.

Response. Thanks for your kindly suggestion. We have revised the title as “Contrasting nidification behaviors facilitate diversification and colonization of the Music frogs under a changing paleoclimate”.

Comment 2. Page 3 line 63. Please replace “nest building” with “nest building (nidification)” to introduce the term for the first time in the text. This would facilitate the reader who heard the term nidification for the first time to learn about this term.

Response. Thanks for your kindly suggestion. We have added “(also known as nidification)” following your comment (line 65 in the marked-up version; line 62 in the clean version).

Comment 3. Page 5 line 99. Please replace “all described species of the genus” with “18 of 20 described species of the genus”. The two newly described species (*Nidirana chongqinensis* and *N. noadihing*) were not included in this study. I understand that by the time the author started their study, these two species were not described yet. Therefore the authors didn’t include these two species. However, since there has been two additional described species, updating the sentence in the text would be recommended.

Response. Thanks for your kindly suggestion. Yes, when we performed this study, the two species were not yet described so that were not included. We have updated the sentence as “from 18 of 20 described species of the genus” (line 105 in the marked-up version; line 99 in the clean version).

Comment 4. Please be consistent in labelling the samples used in this study. The authors wrote *Nidirana guibeiensis* throughout the text but in the Supplementary Table S1 *Nidirana* “*guibeiensis*”.

Response. Thanks for your kindly suggestion. We have revised the nomenclature in the text as “*N. “guibeiensis”*” to indicate it is a synonymy (lines 128 and 139 in the marked-up version; lines 120 and 131 in the clean version). In the table S1 and figure

S1, we use “*Nidirana leishanensis* (*Nidirana* “*guibeiensis*”)” to indicate it is a synonymy of *Nidirana leishanensis* but previously named as *Nidirana* “*guibeiensis*”.

Comment 5. *While I understand that it might not be the scope of this manuscript, I am wondering if it is possible to somehow elaborate/indicate in one or more sentence(s) how the results from this study could be used to project/estimate future populations of this genus under the climatic factors defined in this study.*

Response. Thanks for your kindly suggestion. We have added a subsection in Discussion about the implication of nidification in the context of future climate change scenario. In addition, we discussed the uncertainty surrounding future populations in the face of such variable situations (line 366–371 in the marked-up version; lines 338–343 in the clean version).

Responses to the comments of Reviewer #2

Comment 1. *The only section that in my opinion needs some strong rewriting is the discussion: while the different aspects of the study are nicely discussed under the three headings, I am missing a general discussion of the overall results in context with other studies, such as comparisons to other animal groups, other behaviours/features evolved due to climate changes etc.*

Response. Thanks for your kindly suggestion. We have added a subsection in Discussion to compare mud nests and foam nests, highlighting their different evolutionary modes and potential adaptation mechanisms (line 357–364 in the marked-up version; lines 329–336 in the clean version).

Comment 2. *I am furthermore missing a general conclusion at the end of the discussion and maybe even a statement regarding the meaning of the results in the lights of current climate change.*

Response. Thanks for your kindly suggestion. We have added a subsection in Discussion to project responses of the nidification species in the face of climate changes as well as associated uncertainties (line 366–371 in the marked-up version; lines 338–343 in the clean version).

Comment 3. *line 33-36: this sentence is very hard to understand and should be*

rewritten in a more comprehensible way

Response. Thanks for your kindly suggestion. This sentence has been rephrased (line 32–36 in the marked-up version; lines 32–35 in the clean version).

Comment 4. *line 59-64: most of the amphibian parental care reviews cited here are rather old. The field has developed substantially and there are several much more recent reviews about this topic*

Response. Thanks for your kindly suggestion. We have added more reference here.

Comment 5. *line 67-68: please provide sources (or a review) for the documentation of mud nests (at least in *Boana* and *Limnonectes*)*

Response. Thanks. Added.

Comment 6. *line 456: species name italic*

Response. Thanks for your correction. We have revised the species name *N. adenopleura* into italic. Sorry for our carelessness.

Comment 7. *line 479: please cite the creators of the ggplot2 package*

Response. Done.

REVIEWERS' COMMENTS:

Reviewer #1 (Remarks to the Author):

At this point, I think I am very pleased with the current version. The authors have incorporated the reviewers comments in this revised version of the manuscript. However, I would like to give a tiny suggestions, as follow:

1. Please write the genus name in full every time it is first mentioned in each paragraph. For example line 165, 177, 191, etc. Please check throughout the manuscript.
2. It is a common practice to add information on the standard ethical guideline(s) for handling animal in research in the method section. I would suggest authors adding this information, i.e., in the "Sampling and Sequencing" section.

Reviewer #2 (Remarks to the Author):

The authors did a really good job answering and implementing the reviewers' suggestions. They wrote a very nice new discussion section and thereby highly improved the manuscript!*

*I found a few minor language errors in this new section, which I marked in the attached PDF-document.

Responses to the comments of Reviewer #1

Comment 1. Please write the genus name in full every time it is first mentioned in each paragraph. For example line 165, 177, 191, etc. Please check throughout the manuscript.

Response. Thanks for your kindly suggestion. We have checked throughout the manuscript and revised following your suggestion. Please see the changes with track in the latest manuscript.

Comment 2. It is a common practice to add information on the standard ethical guideline(s) for handling animal in research in the method section. I would suggest authors adding this information, i.e., in the "Sampling and Sequencing" section.

Response. Thanks for your reminder. We have included an ethical declaration within the "Sampling and Sequencing" section (line 362–364).

Responses to the comments of Reviewer #2

Comment 1. The authors did a really good job answering and implementing the reviewers' suggestions. They wrote a very nice new discussion section and thereby highly improved the manuscript! I found a few minor language errors in this new section, which I marked in the attached PDF-document.

Response: Thanks for your kindly edits. The language errors have been thoroughly revised. Please see the changes with track in the latest manuscript.